# A marmoset model for *Mycobacterium avium* complex pulmonary disease

**Jay Peters**[1,2]*, **Diego Jose Maselli**[1,2], **Mandeep Mangat**[1], **Jacqueline J. Coalson**[1],
**Cecilia Hinojosa**[1], **Luis Giavedoni**[2], **Barbara A. Brown-Elliott**[3], **Edward Chan**[4],
**David Griffith**[4,5]

**1** Department of Medicine, University of Texas Health Science Center, San Antonio, Texas, United States of
America, **2** Texas Biomedical Research Institute, San Antonio, Texas, United States of America,
**3** Department of Microbiology, University of Texas Health Science Center, Tyler, TX, United States of
America, **4** Department of Medicine, National Jewish Health, Denver, CO, United States of America,
**5** Department of Medicine, University of Texas Health Science Center, Tyler, TX, United States of America

* PETERS@UTHSCSA.edu

**Data Availability Statement:** All relevant data are located at Dryad (doi:10.5061/dryad.jdfn2z3cr).

**Funding:** Harry L. Willett Foundation, Denver, CO. The funders had no role in study design, data

## Abstract

### Rationale

*Mycobacterium avium* complex, is the most common nontuberculous mycobacterial respiratory pathogen in humans. Disease mechanisms are poorly understood due to the absence of a reliable animal model for *M. avium* complex pulmonary disease.

### Objectives

The objectives of this study were to **a**ssess the susceptibility, immunologic and histopathologic responses of the common marmoset (*Callithrix jacchus*) to *M. avium* complex pulmonary infection.

### Methods

7 adult female marmosets underwent endobronchial inoculation with $10^8$ colony-forming units *of M. intracellulare* and were monitored for 30 or 60 days. Chest radiograph was assessed at baseline (prior to infection) and at the time of sacrifice (30 days for 3 animals and 60 days for 4 animals), and bronchoalveolar lavage cytokines, histopathology and cultures of the bronchoalveolar lavage, lungs, liver and kidney were assessed at time of sacrifice. Serum cytokines were monitored at baseline and weekly for 30 days for all animals and at 60 days for those alive. Group differences in serum cytokine measurements between those that tested positive versus negative for the *M. intracellulare* infection were assessed using a series of linear mixed models.

### Measurements and main results

Five of seven animals (two at 30 days and three at 60 days of infection) had positive lung cultures for *M. intracellulare*. Extra-pulmonary cultures were positive in three animals. All animals appeared healthy throughout the study. All five animals with positive lung cultures had radiographic changes consistent with pneumonitis. At 30 days, those with *M.*

collection and analysis, decision to publish, or
preparation of the manuscript. The authors
received no specific funding for this work.

**Competing interests:** The author have declared no
competing interest exist.

*intracellulare* lung infection showed granulomatous inflammation, while at 60 days there
were fewer inflammatory changes but bronchiectasis was noted. The cytokine response in
the bronchoalveolar lavage fluid was uniformly greater in the animals with positive *M. intra-
cellulare* cultures than those without a productive infection, with greater levels at 30-days
compared to 60-days. Similarly, serum cytokines were more elevated in the animals that
had positive *M. intracellulare* cultures compared to those without a productive infection,
peaking 14–21 days after inoculation.

## Conclusion

Endobronchial instillation of *M. intracellulare* resulted in pulmonary mycobacterial infection
in marmosets with a differential immune response, radiographic and histopathologic abnor-
malities, and an indolent course consistent with *M. avium* complex lung infection in humans.

## Introduction

*Mycobacterium avium* complex (MAC) is the most common nontuberculous mycobacterial
(NTM) respiratory pathogen in humans [1, 2]. MAC comprises multiple species and subspe-
cies including *M. avium* and *M. intracellulare*, the two most important MAC respiratory path-
ogens [3–5]. Typically, these two species are both reported as "MAC," but their environmental
sources differ, and there is evidence indicating differential pathogenicity and clinical disease
severity between the two species [6, 7]. Since MAC is ubiquitous in the environment and expo-
sure is likely unavoidable, it is apparent that some form of host susceptibility must also be pres-
ent for MAC lung disease to occur [3, 4, 8]. In that context, pulmonary MAC disease occurs
primarily in patients with structural lung disease, especially bronchiectasis and emphysema,
without demonstrable systemic immune suppression [3, 4, 8]. Pathophysiologic questions are
further complicated because MAC lung disease can evolve in two forms, either fibro-cavitary
disease similarly to pulmonary tuberculosis (TB), or as a more indolent infection associated
radiographically with nodules and bronchiectasis (nodular/bronchiectatic disease) [3–5].

A major impediment to greater understanding of fundamental pathophysiologic mecha-
nisms surrounding MAC lung disease is the lack of a reproducible animal model that can repli-
cate cellular, biochemical and pathological events observed in human MAC lung disease. The
relevance of murine models of MAC infection for human MAC lung disease is not established
and has uncertain applicability [9]. Presumably, MAC lung infection, in a host species more
related to humans, would be more informative and has been achieved with endobronchial
instillation of *M. avium* in a rhesus macaque [10]. While it also remains unclear if the mecha-
nisms of disease establishment and progression in this model are pertinent to human MAC
lung disease [10], an animal model using a species phylogenetically even closer to humans
should provide a better approximation to the human mycobacterial response.

The common marmoset (*Callithrix jacchus*) has been used as a model for TB lung infection
[9–12]. Endotracheal instillation of TB strains in marmosets results in pulmonary abnormali-
ties covering the entire spectrum of lesions observed in human TB patients including cavita-
tion [11–14]. However, the utility of marmosets as a model for human TB disease is limited
because marmosets are exceptionally susceptible to *M. tuberculosis* [11–14]. This susceptibility
to mycobacterial infection combined with pathophysiologic similarities between marmosets
and humans to *M. tuberculosis* infection are potential advantages for utilizing marmosets to

study infection with a relatively non-virulent human mycobacterial pathogen such as MAC. Additionally, marmosets are smaller than macaques with relatively shorter life span, allowing feasible studies into advanced age. For these reasons, we investigated the potential of marmosets as a model for MAC lung disease.

## Methods

### Non-human primate research regulations

Seven adult colony-bred female marmosets (*Callithrix jacchus*) were purchased from the Southwest National Primate Center, San Antonio, TX, and were found to be free of known primate bacterial and viral pathogens based on routine surveillance. The study and use of non-human primates were conducted in accordance with the Guidelines established by the Weatherall report and conformed to National Institutes of Health guidelines [15, 16]. All animal work was approved by the University of Texas Health Science Center, San Antonio (UTHSC-SA) and the University of Texas Health Science Center, Tyler (UTHSCT) Institutional Animal Care and Use Committees and the Southwest National Primate Research Center Institutional Animal Care and Use Committee of the Texas Biomedical Research Institute (Texas Biomed). Animals were housed separately and had vital signs (heart rate, temperature, and oxygen saturation) monitored closely throughout the experimental period. They were weighed daily, measured for nutritional and fluid intake, and examined twice daily for normal interactions with staff members.

### Animal infection and sample collection

The seven adult marmosets were inoculated endobronchially at the level of the main carina using a special narrow diameter bronchoscope with one mL of a $10^8$ CFU/mL *M. intracellulare* obtained from the Mycobacteria/Nocardia Research Laboratory at the UTHSCT. All procedures (bronchoscopy, blood draws and euthanasia) were conducted under ketamine anesthesia with the additional use of isoflurane anesthesia with bronchoscopy and bronchoalveolar lavage (BAL) in the presence of veterinary staff. Each animal underwent assessment of serum chemistry, and complete blood count prior to inoculation and on the day of euthanasia. Because there are no previous comparable studies with this primate, we sacrificed a group of animals at 30 days and another group at 60 days to optimize the chance of recovering *M. intracelluare* as well as to define the time course of an evolving inflammatory response. Cytokine analysis was obtained prior to inoculation with *M. intracellualre* and on a weekly basis from day 0 to day 30 for all animals and again on day 60 for the animals sacrificed at day 60. All the animals had BAL performed prior to euthanasia at either 30- or 60-days post-inoculation. The animals were then taken directly to necropsy by a primate pathologist.

### Imaging

Supine postero-anterior and lateral chest X-rays were obtained at baseline and prior to sacrifice.

### Histopathology

Formalin-fixed, paraffin-embedded tissue sections were deparaffinized and stained with hematoxylin and eosin for histopathological analysis as well as for acid-fast bacteria (AFB) stain. Histopathologic images were obtained using an Axioplan microscope (Carl Zeiss, Jena, Germany) with a Spot Insight camera (Diagnostic Instruments Inc., Sterling Heights, MI).

## Microbiologic assessments

A macrolide- and aminoglycoside-resistant (clarithromycin MIC >16 μg/mL and amikacin MIC >64 μg/mL) isolate of *M. intracellulare*, previously identified by 16S rRNA gene sequence, was prepared for inoculating the marmosets. The clinical isolate was grown on Middlebrook 7H10 agar. After adequate growth was obtained (approximately 7–10 days), several colonies were transferred to 3 mL of sterile distilled water to prepare a suspension with optical density equal to a 0.5 McFarland standard by nephelometer reading. The inoculum was chosen since this turbidity represents the approximate number of organisms ($10^8$ CFU/mL) present in the matched turbidity McFarland standard used for antimicrobial susceptibility testing as recommended by the Clinical and Laboratory Standards Institute (CLSI) [17]. The suspension was incubated for 7 days at 35°C and 1–3 mL aliquots prepared to be used to inoculate the marmosets.

BAL and tissue samples were processed and cultured for mycobacteria by the Mycobacteria/Nocardia Research Laboratory at the UTHSCT, using standard decontamination procedures, fluorochrome microscopy, solid media culture on a biplate of Middlebrook 7H10 agar with and without antibiotics, and a broth culture (BACTEC 960, Becton Dickinson and Company, Sparks, MD, VersaTrek, Thermofisher, formerly Trek Diagnostic Systems, Cleveland, Ohio) as previously described [18]. *M. intracullulare* isolates were identified using AccuProbe (Hologic-GenProbe, San Diego, CA, as previously described [18]. *In vitro* susceptibility testing of MAC isolates was performed as previously described [17]. *M. intracellulare* growth on broth and solid media was assessed using semi-quantitative scoring: growth on broth medium only = "pos", growth in broth medium plus 1–49 countable colonies (cc) on solid medium, 50–99 cc on solid medium = 1+, 100–199 cc on solid medium = 2+, 200–299 cc on solid medium = 3 +, greater than 300 cc on solid medium = 4+ [19].

## Cytokine analysis

Batched BAL supernatant and plasma samples were stored at -80°C. They were then collectively thawed and analyzed in duplicate using the Invitrogen Cytokine Monkey Magnetic 28-plex Panel which includes monocyte chemo-attractant protein 1 (MCP-1), interleukin-12 (IL-12), granulocyte-monocyte colony stimulating factor (GM-CSF), macrophage inflammatory protein 1 beta (MIP 1-β), interferon-gamma (IFNγ), monokine induced by interferon-gamma (MIG), migration inhibition factor (MIF), IL-1 receptor antagonist (IL-1Ra), tumor necrosis factor (TNF), IL-2, IL-4, IL-8, intercellular adhesion molecule (ICAM), and RANTES (Regulated on Activation, normal T cell Expressed and Secreted). For animals sacrificed at day 30, plasma samples were obtained at baseline and on days 7, 14, 21, 30 with BAL samples collected at day 30. For animals sacrificed at day 60, plasma samples were collected at baseline and on days 7, 14, 21, 28, and 60 along with BAL samples collected at day 60.

## Tissue analysis

At the time of sacrifice, fresh tissues from the mediastinal lymph nodes, liver, spleen, and kidneys were prepared for culture. The lungs were resected *en-bloc* and inspected visually for areas of inflammation. One lobe that appeared abnormal was isolated, tied off, and resected for culture. The remaining segments of lung were suspended after cannulating the trachea and then inflated with formalin infused at a height of 30 centimeters. After inflation, the tracheal was tied off and the lungs; as well as sections from the liver, spleen, and kidneys were submerged in 100% formalin for a period of 10 days. All tissues were then sent to the Central Pathology Lab at the UTHSC-SA for processing and staining.

## Statistical analysis

All cytokine measurements were log-transformed. Cytokine measurements below the limit of detection were imputed using $\frac{1}{\sqrt{2}} LDL_{cytokine}$, where $LDL_{cytokine}$ is the lower limit of detection for a given cytokine and has a value of 1 for all cytokines for the assay used. Longitudinal trajectory plots were generated for all log-transformed cytokines to assess the curvilinear nature of cytokine response through time. A series of linear mixed effects models were utilized to assess differences in log-transformed cytokine measurements between animals that tested positive versus negative for *M. intracellulare* infection. All models included a term for group (+ve vs. -ve for *M. intracellulare* infection), time, time$^2$ and time$^3$ variables, a random intercept term for animal, and an unstructured error covariance structure to account for correlation between repeated measurements from the same animal. Time, measured in weeks, was treated as a continuous variable in all models. The quadratic and cubic terms for time were included based on initial assessment of the cytokine trajectory plots.

## Results

### Clinical and standard laboratory assessments of the *M. intracellulare*-infected marmosets

All animals had normal growth and behavior as well as normal blood chemistries and cellular counts prior to infection with *M. intracellulare*. Following instillation of *M. intracellulare*, the animals were regularly assessed for evidence of illness as per protocol for Texas Biomed. Throughout the study period, none of the animals displayed signs of respiratory disease such as cough or tachypnea. Infected animals initially lost weight during the first two weeks but returned toward baseline by the day of sacrifice (Table 1). All the marmosets exhibited normal behavior and activity until the time of sacrifice.

### Mycobacterial burden

For the three marmosets that were sacrificed at day 30, two were culture positive for *M. intracellulare* in the lungs and spleens, and one was culture negative for the mycobacteria in all organs (Table 1). Of the two animals with lung culture positivity, one also had positive *M. intracellulare* cultures in the spleen and kidneys and the other also had positive culture in the spleen. Interestingly, all three animals had negative BAL cultures at day 30 just prior to euthanasia. No specimen submitted for AFB stain (smear) and culture was AFB stain positive.

For the four marmosets that were sacrificed at day 60, three had positive *M. intracellulare* cultures of the lungs and one was negative for the mycobacteria in all organs (Table 1). Of the three animals with positive *M. intracellulare* lung cultures, only one had a positive extra-pulmonary culture in the spleen and liver. Similar to the marmosets infected for 30 days, all had negative BAL cultures for mycobacteria at day 60. No specimen submitted for AFB culture was AFB stain positive, although some tissue specimens were AFB smear positive on histopathologic analysis.

On semi-quantitative analysis, all positive AFB cultures were scored "countable colonies", 1–49 colonies on solid media) with two exceptions (Table 1). There was no apparent correlation between the degree of culture positivity from the lungs and the presence of positive extra-pulmonary cultures. All positive *M. intracellulare* cultures had the identical *in vitro* susceptibility pattern as the originally instilled *M. intracellulare* isolate with macrolide and amikacin resistance. Staining with fluorochrome confirmed the presence of AFB in the *M. intracellulare*-infected lung tissue samples (Fig 1).

**Table 1. Summary of microbiologic, radiographic and pathologic findings from each animal.**

| Animal (wt-pre/post, gms) | Day sacrificed | M. intracellulare AFB culture[1,2] | | | | | CXR | Path |
|---|---|---|---|---|---|---|---|---|
| | | **Lung** | **Spleen** | **Kidney** | **Liver** | **BAL** | | |
| 1 (474/473) | 30 | cc | cc | cc | – | – | + | + |
| | | | | | | | | |
| 2 (390/388) | 30 | 3+ | cc | – | – | – | + | + |
| | | | | | | | | |
| 3 (480/481) | 30 | – | – | – | – | – | – | – |
| | | - | - | - | - | | | |
| 4 (500/497) | 60 | cc | – | – | – | – | + | + |
| | | | | | | | | |
| 5 (401/408) | 60 | | cc | – | pos | – | + | + |
| | | cc | | | | | | |
| 6 (486/472) | 60 | cc | – | – | – | – | + | + |
| | | | | - | - | | | |
| 7 (475/473) | 60 | – | – | – | – | – | – | – |
| | | - | - | - | - | | | |

| **Chest X-ray**[3] | **Lung histopathology**[3] |
|---|---|
| 1: RLL posterior consolidation | 1: RML, RLL, Left lung: diffuse involvement with lymphocytes & monocytes. Early granulomatous inflammation, few giant cells. Subpleural and mediastinal subscapular inflammation |
| 2: RLL posterior consolidation | 2: Right lung 60% consolidated, mixed lymphocytes, monocytes, numerous granulomas and giant cells, significant edema around lymphatics. Left lung 10% consolidation, similar findings |
| 3. Unremarkable | 3: RML, RLL, LLL: normal |
| 4. RLL posterior consolidation | 4: RUL, RLL, LLL: Resolving inflammation with lymphocytes /monocytes worse in subpleural region; no granulomas; RML airway damage extending to the pleural surface with bronchiectasis |
| 5. RLL posterior consolidation | 5: RUL, RLL, LLL: Mild/moderate inflammation with lymphocytes/ monocytes, and some PMNs RLL |
| 6. RLL posterior consolidation | 6: RUL, RLL, LLL: Mild/moderate inflammation with lymph0cytes/ monocytes, RLL with early bronchiectasis |
| 7. Unremarkable | 7: RML, RLL, LLL: Normal |

Change in weight, microbiology, pathology and radiology: (1) M. intracellulare growth on broth and solid media: growth on broth medium only = "pos", growth in broth medium plus 1–49 countable colonies (cc)on solid medium, 50–99 cc on solid medium = 1+, 100–199 cc on solid medium = 2+, 200–299 cc on solid medium = 3 +, greater than 300 cc on solid medium = 4+ (Ref 19). (2)No specimens that were submitted for AFB stain and culture to the mycobacteriology laboratory had a positive AFB stain; however, some lung specimens submitted for histopathologic analysis were AFB stain positive. (3)RUL = right upper lobe, RML = right middle lobe, RLL = right lower lobe, LUL = left upper lobe, LLL = left lower lobe

## Chest radiographic features

The chest radiographs of all seven marmosets were within normal limits prior to infection with *M. intracellulare*. Following infection, the chest radiographs were abnormal with evidence of patchy consolidation in 5/5 animals with lung cultures positive for *M. intracellulare* (Fig 2). The animal with the culture score of "5" on semi-quantitative analysis also had the most extensive radiographic abnormalities (Table 1, Fig 1). The two animals with negative lung cultures for *M. intracelluare* had no end of study chest radiographic abnormalities.

## Histopathologic findings

For the 5 infected animals, all five had visible hemorrhagic abnormalities in the lungs. Histopathologic features in the lungs of animals with culturable *M. intracelluare* at day 30 included mixed monocytic and lymphocytic infiltration around the bronchovascular bundle extending into the alveolar space. Numerous early granulomas with associated giant cells and occasional neutrophils and eosinophils were observed (Fig 3). Subpleural intra-alveolar nodules with

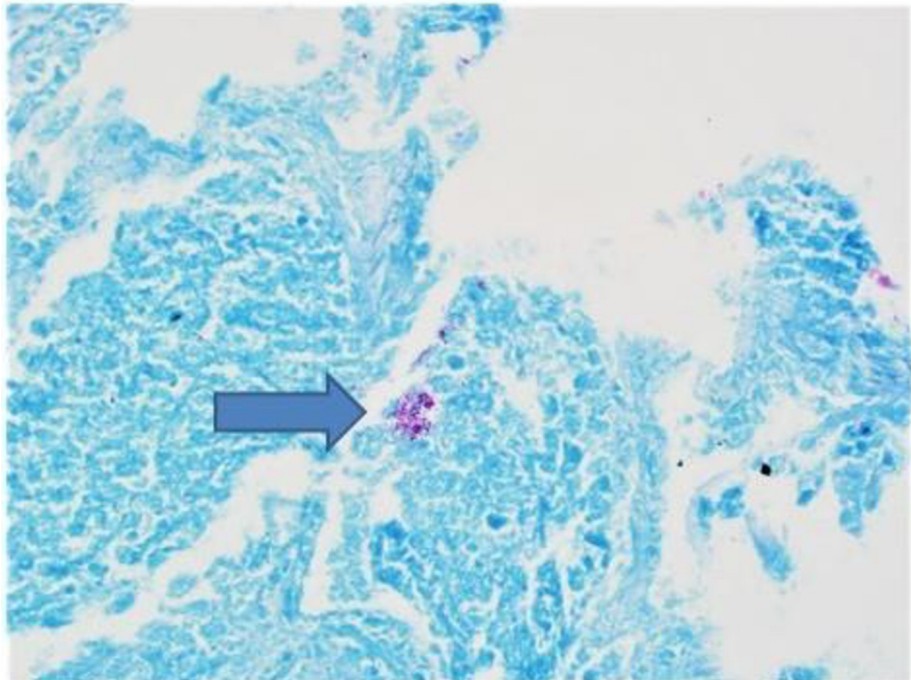

**Fig 1. Granulomatous inflammation with positive AFB stain.** Histopathologic examination from marmoset #2 that was sacrificed 30 days after inoculation with M. intracellulare. Most of the right lung was consolidated with mixed lymphocytes, monocytes, and numerous granulomas. Occasional AFB (noted by arrow) were identified in the areas of granulomatous inflammation.

severe pleuritis were seen in three animals. One animal had marked enlargement of the mediastinal lymph nodes with striking subcapsular mediastinal lymphadenitis. At day 60, animals were noted to have resolving inflammation in the lung parenchyma with residual bronchitis and bronchiolitis. Two animals had findings consistent with early bronchiectasis. Minimal or no granulomatous inflammation was noted by day 60. The spleen, kidney, and liver histologic sections from the animals with positive cultures from those organs did not show abnormalities including evidence of inflammation typical of mycobacterial infection.

## BAL cytokines

Cytokine and chemokine levels were also quantified in the BAL fluids of all animals prior to sacrifice at either day 30 or day 60. To varying degrees, BAL cytokine levels from the animals with positive lung cultures at either the day 30 or day 60 of infection (total n = 5) had increased levels of MIF, MIP-1α, MIP-1β, IL-1Ra, MIG, ICAM, IFNγ, RANTES, and TNF, compared to animals without a productive infection (total n = 2). BAL levels for all the cytokines and chemokines were greater at day 30 except for MIF, which was greater at day 60. BAL cytokine levels tracked serum cytokines but remained elevated to day 60 (see below). Cytokines and chemokines in the 28-cytokine kit not mentioned were below the detection limit for both the serum and BAL assays.

## Cytokines

Seven IFN-g and nine MIG measurements below the limit of detection were imputed as per protocol. Compared to animals that tested negative for the *M. intracellulate* infection, cytokine measurements were consistently significantly higher in animals that tested positive for the

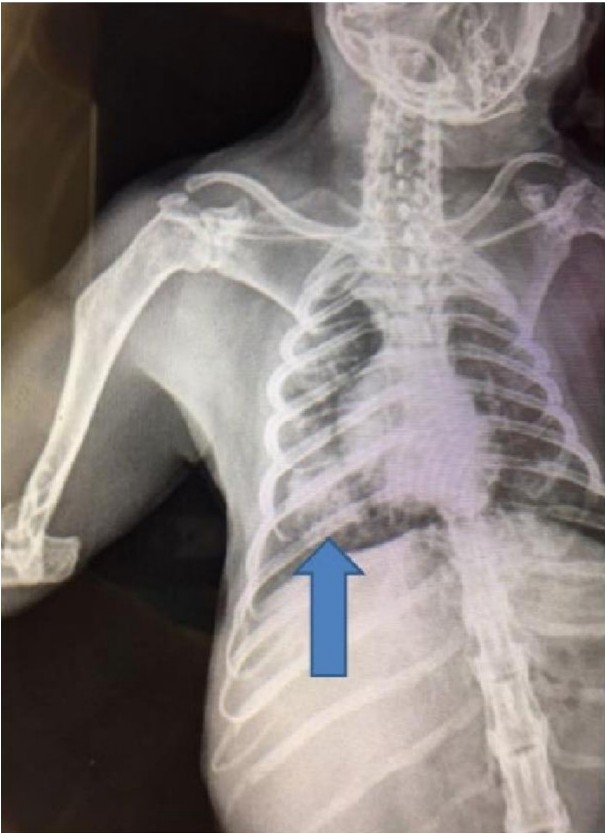

**Fig 2. Chest radiograph prior to necropsy from marmoset #2 showing a superior segment right lower lobe infiltrate.** Right lower lobe consolidation with numerous granulomas were confirmed on histologic exam.

infection ($p < 0.05$; Table 2). Mean log-cytokine measurements for animals that tested positive were 1.42, 1.18, 0.99, 0.51, 0.32 and 0.26 units higher for IFN-g, MIG, MIF, MIP-1a, IL-1Ra and MIP-1b, respectively (Table 2). We also observed significant cubic relationship with time for IFN-g, MIF, MIP-1a and MIP-1b, and a significant quadratic relationship with time for IL-1Ra. For all cytokines, increase in response continued for approximately 14 to 21 days, followed by some decline for both groups (Fig 4).

## Discussion

In this study, we successfully established pulmonary *M. intracellulare* infection in five of the seven healthy female marmosets. We chose female marmosets for this investigation due to the predominance of human *M. avium* and *M. intracellulare* lung disease in women [3–5]. The animals showed no discernable clinical signs of lung infection, such as cough, decreased activity or weight loss, suggesting that the inoculation resulted in a "sub-clinical" infection. While there was culture positivity in extra-pulmonary organs in a few animals, there were no accompanying histopathologic findings. This type of indolent infection is consistent with the nodular/bronchiectasis form of *M. intracellulare* lung disease in humans [3–5].

All *M. intracellulare* isolates cultured from the infected animals showed the same *in vitro* susceptibility pattern as the *M. intracellulare* isolate instilled in the animals, specifically, *in vitro* resistance to macrolide and amikacin, precluding the possibility of environmental *M. intracellulare* contamination of the tissue (and BAL) specimens. Two animals, one sacrificed at

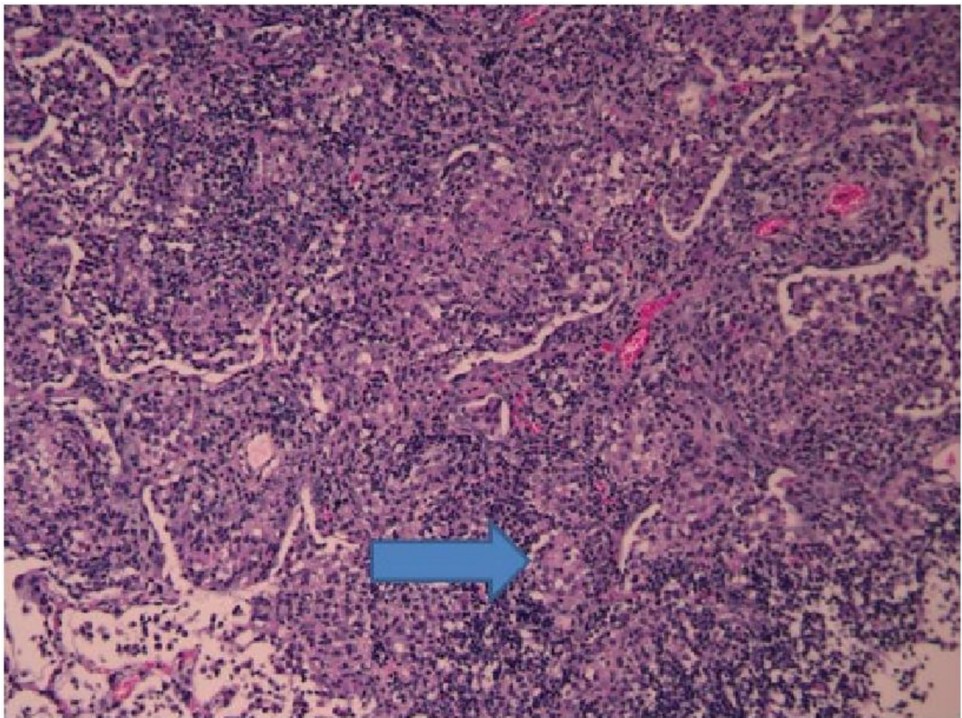

**Fig 3. Animal #2, day 30 histology taken from the right middle lobe showing early granuloma with giant cells, mixed lymphocytic/monocytic infiltration with occasional neutrophils and eosinophils.** Similar findings with numerous granulomas were identified in much of the right lower lobe.

30 days and one at 60 days, did not show evidence of infection. For both of these animals, there was *a priori* a question about the adequacy of the endotracheal *M. intracellulare* challenge with technical problems resulting in a reduction in the amount of inoculum delivered.

The serum and BAL cytokine and chemokine levels for animals sacrificed at 30 and 60 days showed variability between animals but there were consistently higher levels for five animals with a productive *M. intracellulare* infection compared to the two animals without recoverable *M. intracelluare* from any of the organs.

Comparing serum and BAL cytokine and chemokine levels at both 30 days and 60 days for each cytokine, two general findings are that the serum levels peaked approximately day 14–30 that tapered off by day 60; whereas, the BAL cytokine and chemokine levels remained elevated in the BAL even at day 60 in the *M. intracellulare* infected animals, suggesting a stronger local effect of the infection than a systemic one. Although there were consistent trends in cytokine/ chemokine response to infection there was considerable variability between animals in the magnitude of response.

Our findings are consistent with those published for other mycobacterial diseases [20]. MIP-1 α/β (now known as CCL3/CCL4)–members of the C-C superfamily associated with the early immune response–peaked at day 14 in the *M. intracellulare*-infected animals. This may account for the mixed monocytic/neutrophilic response seen in our model. IFNγ, considered to be essential for the induction of granulomatous inflammation, peaked between days 14–21. This was associated with a rise in MIF, known to be induced by IFNγ, which peaked between days 14–30. MIF is felt to contribute to detrimental inflammation but may be crucial in controlling infection caused by mycobacterial diseases [21, 22]. RANTES (CCL-5), which promotes macrophage chemotaxis and upregulation, peaked between days 30–60. High levels of

**Table 2. Temporal change in cytokine expression.**

| Cytokine | Predictor term | Mean change | 95% CI | P-value |
|---|---|---|---|---|
| IFN-γ | Group:Positive | 1.42 | (1.14 to 1.73) | <0.001 |
| | Time | 1.15 | (0.73 to 1.57) | <0.001 |
| | Time$^2$ | -0.26 | (-0.41 to -0.11) | 0.002 |
| | Time$^3$ | 0.02 | (0.004 to 0.03) | 0.018 |
| MIG | Group:Positive | 1.18 | (0.80 to 1.55) | <0.001 |
| | Time | 0.55 | (0.01 to 1.08) | 0.064 |
| | Time$^2$ | -0.05 | (-0.24 to 0.13) | 0.611 |
| | Time$^3$ | -0.0002 | (-0.02 to 0.01) | 0.985 |
| MIF | Group:Positive | 0.99 | (0.27 to 1.69) | 0.045 |
| | Time | 0.80 | (0.53 to 1.07) | <0.001 |
| | Time$^2$ | -0.21 | (-0.31 to -0.12) | <0.001 |
| | Time$^3$ | 0.01 | (0.01 to 0.02) | <0.001 |
| MIP-1α | Group:Positive | 0.51 | (0.36 to 0.66) | 0.001 |
| | Time | 0.40 | (0.21 to 0.58) | <0.001 |
| | Time$^2$ | -0.11 | (-0.17 to -0.05) | 0.002 |
| | Time$^3$ | 0.01 | (0.002 to 0.01) | 0.009 |
| IL-1Ra | Group:Positive | 0.32 | (0.11 to 0.53) | 0.031 |
| | Time | 0.36 | (0.13 to 0.60) | 0.006 |
| | Time$^2$ | -0.09 | (-0.17 to -0.01) | 0.047 |
| | Time$^3$ | 0.01 | (-0.001 to 0.01) | 0.134 |
| MIP-1β | Group:Positive | 0.26 | (0.12 to 0.40) | 0.016 |
| | Time | 0.44 | (0.24 to 0.63) | <0.001 |
| | Time$^2$ | -0.11 | (-0.18 to -0.05) | 0.004 |
| | Time$^3$ | 0.01 | (0.002 to 0.01) | 0.016 |

**IFN**-γ = interferon-gamma; **MIG** = monokine induced by gamma-interferon; **MIF** = migration inhibitory factor

**MIP-1**α = macrophage inflammatory protein-1-alpha; **IL-1Ra** = interleukin-1 receptor antagonist

**MIP-1**β = macrophage inflammatory protein-1-beta.

Changes in log-transformed cytokine measurements, 95% CI and p-value based on a series of linear mixed models including log-transformed cytokine as a response variable and predictor terms for group, time (measured in weeks), time2 and time3 as fixed effects. Group: Positive indicates mean change in animals that tested positive for M. intracellulare infection relative to animals that tested negative for the infection.

TNF, the only serum cytokine not persistently higher at day 60, has been associated with necrosis in animal models of tuberculosis and was not observed in our model. As noted, although there were consistent trends in cytokine/chemokine response to infection in our study, there was considerable variability between animals in the magnitude of response.

Histopathological analysis of the *M. intracellulare* culture-positive lungs showed typical abnormalities consistent with granulomatous inflammation with giant cells but no evidence of necrosis, the latter consistent with the absence of TNF. Stains of lung tissue were inconsistently AFB smear positive yet consistently culture positive for *M. intracellulare*, similar to findings in patients with MAC lung disease [3–5]. Additionally, by day 60, the lungs showed resolving inflammation with persistent bronchitis and bronchiolitis and early evidence of bronchiectasis. These latter findings, in the context of persistently positive MAC cultures, would be consistent with chronic mycobacterial lung infections in humans.

Fibro-cavitary forms of MAC lung infection, including *M. intracellulare*, are clearly associated with bronchiectasis formation, but it has never been established in the nodular/bronchiectasis form of MAC lung disease, whether MAC can initiate bronchiectasis formation [23]. The current consensus is that bronchiectasis likely precedes MAC infection for most patients and

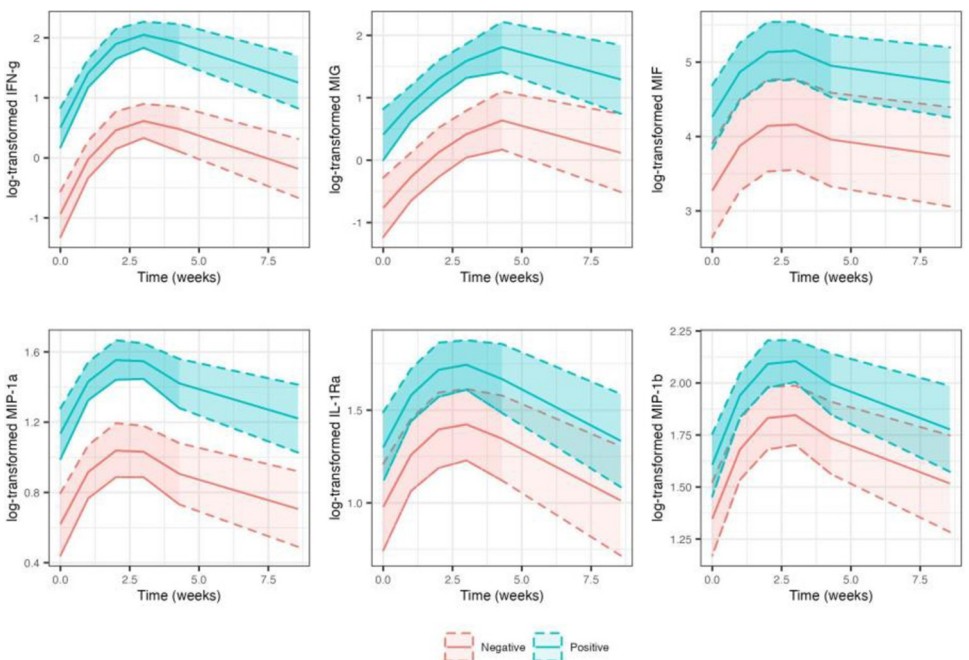

**Fig 4. Predicted mean log-transformed cytokine response and 95% confidence band by time (measured in weeks) in animals that tested positive (blue line) versus negative (red line) for M. intracellulare infection based on linear mixed model fits.**

is, in fact, the critical predisposition for the establishment of MAC lung infection [3–5]. Our results suggest that the MAC infection per se has the potential to cause bronchiectasis formation. Another possibility is that if some degree of pre-existing bronchiectasis is present prior to the establishment of MAC lung infection the bronchiectasis may contribute to or accelerate further bronchiectasis formation.

Winthrop et al. published results with three rhesus macaques infected with escalating doses ($10^6$, $10^7$, $10^8$ CFU) of *M. avium* subsp. *hominissuis* strain 101 administered endobronchially [10]. The animals that received $10^9$ CFU *M. avium* developed a right lung opacity radiographically on day 14 post-infection. The radiographic abnormality was associated with recovery of *M. avium* in BAL fluid culture. Similar to our findings, there were mid-infection increases in circulating cytokines, IL-6, IL-12 and IFNγ with peak levels at day 14 post- inoculation for IL-12, day 21 for IFNγ and day 28 for IL-6. BAL cytokines that were elevated peaked at mid-infection (42 days post inoculation) included, IL-6, IL-12, IFNγ, TNF, MIP-1β. Interestingly, only IL-6 did not return to baseline at the end of the infection period.

Our findings, from a pathophysiologic perspective, are generally consistent with the findings from Winthrop et al. [10]. It is possible that some dissimilarities could be due to differences in virulence and pathogenicity between *M. intracellulare* and *M. avium* (as has been noted in humans), differences in disease pathophysiology between the rhesus macaque and marmoset, or both [6, 7].

A novel finding in this animal model was the positive cultures for *M. intracellulare* in extra-pulmonary organs including spleen, liver and kidney, without apparent clinically detectable infection in the extra-pulmonary organs as well as an absence of discernable inflammatory response in the involved organs. It is possible that the apparent dissemination of the organism is related to the relative susceptibility of marmosets to mycobacteria, although in contrast to TB disease in marmosets, the *M. intracellulare* infected animals did not die or even develop

clinical signs of disease with the dissemination. Alternatively, *M. intracellulare* lung infection may be pathophysiologically similarly to a latent TB state with early dissemination followed by immune-mediated control of the infection in extra-pulmonary organs. The emergence of disseminated *M. avium* complex disease in individuals with advanced AIDS would lend credence to this notion of a latent NTM infection with subsequent reactivation with development of an immunocompromised state.

We cannot confidently assert that any current animal model faithfully replicates the pathophysiologic events associated with human MAC lung disease. With the current level of knowledge, there is no reliable way to correlate the animal findings of early and evolving MAC infection with those from humans, which are largely unknown. While this study does not present a comprehensive description of MAC lung disease pathophysiology, we believe that potential exists with the marmoset MAC lung disease model to accomplish this goal.

One important potential advantage of marmosets for studying *M. intracellulare* or *M. avium* lung disease is that they are relatively susceptible to NTM lung infection so that establishment of *M. intracellulare* lung infection does not require an immune compromised state or airway injury. Based on our success in establishing *M. intracellulare* infection in 5/7 challenged animals, marmosets appear to be a reproducible model for establishing *M. intracellulare* lung infection which could impact several areas of investigation. First, this model could more rigorously identify mechanisms of host susceptibility and disease progression pertinent to humans. For example, marmosets could help identify virulence differences between *M. intracellulare* and *M. avium*, and might also serve as a model for less common NTM pathogens such as *M. abscessus* or *M. xenopi*. Second, a reliable non-human primate model of MAC lung infection could also prove more informative and predictive of drug responses than current non-primate models. Third, the development of bronchiectasis in the marmosets, even after a short follow-up time, could provide insights into the pathophysiology of this complex process. Fourth, translational studies comparing the immune response in the lung tissues of MAC-infected marmosets with that of surgically removed lung tissues of MAC lung disease patients may provide insights into the pathogenesis of progressive NTM lung disease in humans.

There are several limitations to this study. Although significant group differences were observed in cytokine response between animals that tested positive verses negative for the *M. intracellulare* infection, we acknowledge the limitation posed by small sample sizes in both groups, especially at the later time points. As such, the analyses performed here are exploratory and need to be verified by further studies. We also did not assess cellular activity by BAL prior to and after challenge with M. intracellulare. The marmoset is a small non-human primate and we were unsure of the safety of repetitive bronchoscopy and BAL. Additionally, we did not demonstrate a dose-response to MAC in this model. While this would have strengthened the model, a larger number of animals would have been required and was too expensive for an initial study. The inoculum of MAC was based on the dose causing infection in the rhesus macaque [10]. In this study by Winthrop et. al., three oophorectomized female macaques were exposed to $10^6$, $10^7$, or $10^8$ of MAC. Only the animal exposed to $10^8$ of MAC developed radiologic and histologic infection. The fact that the dose of inoculum at $10^8$ was appropriate is supported by the observation that two of the animals in this study received a limited amount of inoculum and neither developed radiographic or histologic changes. Additionally, we injected $10^8$ of M. Abscessus into marmosets (two at 30 days; 1 at 60 days) and none of the animals developed any radiologic or histologic changes (data not shown). Another limitation of our study was an inability to demonstrate the effect of gender in the development of infection. Female marmosets were chosen because MAC disease in humans is predominately seen in women. Chest CT and PET-CT were not obtained and would have been more sensitive for exposing areas of mycobacterial infection.

In conclusion, we demonstrated that endobronchial instillation of *M. intracelluare* reliably results in pulmonary mycobacterial infection in marmosets. The infection is associated with a reproducible immune reaction, radiographic abnormalities and histologic changes consistent with mycobacterial lung infection in humans. Additionally, the infection is indolent without visible acute harm or impact to the animal, similar to the *M. intracellulare*-associated lung disease in humans. We believe this model has the potential to answer questions about *M. intracellulare* disease pathophysiology as well as the natural history of *M. intracellulare* lung disease. To address those questions, future investigations will need to include longer post-exposure observations of the animals and more extensive investigation of pathophysiologic mechanisms.

## Acknowledgments

We are grateful to Lore Fornis for expert help with the graphs and to Dr. Matthew Strand and Aastha Khatiwanda PhD in the Biostatistics Department at National Jewish Health for intellectual input and statistical analysis.

## Author Contributions

**Conceptualization:** Jay Peters, Diego Jose Maselli, Jacqueline J. Coalson, David Griffith.

**Data curation:** Jay Peters, Diego Jose Maselli, Cecilia Hinojosa, Luis Giavedoni, Barbara A. Brown-Elliott, David Griffith.

**Formal analysis:** Jay Peters, Diego Jose Maselli, Jacqueline J. Coalson, Luis Giavedoni, Edward Chan.

**Funding acquisition:** David Griffith.

**Investigation:** Jay Peters, Diego Jose Maselli, Mandeep Mangat, Cecilia Hinojosa, Luis Giavedoni, Barbara A. Brown-Elliott.

**Methodology:** Jay Peters, David Griffith.

**Resources:** Mandeep Mangat.

**Validation:** David Griffith.

**Writing – original draft:** Jay Peters, David Griffith.

**Writing – review & editing:** Jay Peters, Diego Jose Maselli, Mandeep Mangat, Jacqueline J. Coalson, Edward Chan, David Griffith.

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
