## [Decision Letter · Decision Letter 0]

11 Jan 2022

PONE-D-21-34410A Marmoset Model for Mycobacterium avium Complex Pulmonary DiseasePLOS ONE

Dear Dr. Peters,

Thank you for submitting your manuscript to PLOS ONE. After careful consideration, we feel that it has merit but does not fully meet PLOS ONE’s publication criteria as it currently stands. Therefore, we invite you to submit a revised version of the manuscript that addresses the points raised during the review process.

ACADEMIC EDITOR: Although this is an interesting article that can improve our knowledge of preclinical animal models of mycobacterial pathogenesis, there are some issues that needs to be addressed through a major revision. Please refer to the reviewer comments and address their concerns on the manuscript as well as in the rebuttal letter. I suggest the authors to take this opportunity to thoroughly edit/revise the entire manuscript for typographical/grammatical errors and clarity of the figures etc.,==============================

We look forward to receiving your revised manuscript.

Kind regards,

Selvakumar Subbian, Ph.D.

Academic Editor

PLOS ONE

Journal Requirements:

“No The funders had no role in study design, data collection and analysis, decision to publish, or preparation of the manuscript.”

Reviewers' comments:

Reviewer's Responses to Questions

**Comments to the Author**

1. Is the manuscript technically sound, and do the data support the conclusions?

Reviewer #1: Yes

Reviewer #2: No

2. Has the statistical analysis been performed appropriately and rigorously? 

Reviewer #1: Yes

Reviewer #2: No

3. Have the authors made all data underlying the findings in their manuscript fully available?

Reviewer #1: Yes

Reviewer #2: Yes

4. Is the manuscript presented in an intelligible fashion and written in standard English?

Reviewer #1: Yes

Reviewer #2: Yes

5. Review Comments to the Author

Reviewer #1: In this manuscript by Peters, et al., the authors seek to fill a major hole in MAC research- the lack of a tractable animal model. Marmosets have been used recently to model Mtb infection and appear to be quite susceptible to Mtb. Here, authors infected 7 marmosets bronchoscopically with high-dose M. intracellulare and 2/3 were culture-positive at necropsy 30 dpi; 3/4 at necropsy 60 dpi. BAL was culture-negative in all cases. There were signs of pneumonitis and some histopathology in the 5 infected animals, although all appeared clinically normal. Culture-positive animals exhibited higher cytokine levels in BALF and serum than did the uninfected animals. M. intracellulare resulted in a mild disease course in marmosets, consistent with MAC presentation in most humans. One issue with this study is that CXR is quite insensitive for detecting subtle changes in lungs and CT imaging would have much higher resolution. However, the authors do note this in the Discussion and the careful histopathology examination was capable of characterizing lung pathology.

The manuscript from leaders in the MAC field is well-written and the studies were well-designed and executed. Though done with a small number of animals (only 5 animals were apparently infected), this report does demonstrate some utility of the marmoset model and may useful for future studies of MAC. One downside to the model is the inconsistency in establishing infection.

The existing manuscript has no flaws that should preclude publication as-is. Only one minor concern lingers: Cytokines were measured with an Invitrogen 28-plex panel. Such multiplex panels have variable ability to detect cyto/chemokines from various NHP species. Was the ability to detect marmoset cyto/chemokines validated?

Reviewer #2: There are a number of questions and concerns regarding the manuscript:

1) Although it is recognized that the number of animals used in the study was limited, it is not possible to draw conclusions regarding the presence of productive/progressive lung infection in the infected animals based on lung CFU data. The endobronchial instillation of 1E8 CFU is very high and much higher than what a human would be exposed to in the natural environment. Interpretation is complicated by the lack of lung deposition CFU after inoculation which is standard in lung infection experiments. Finally, the choice of scoring positive cultures as countable colonies rather than a more standard criteria such as CFU per gram of lung tissue further clouds the interpretation. Within the countable colony from 1-49 the range is quite large, if 1 or 2 colonies were present this would represent negligible recovery of bacteria from the lung.

2) Minimal information is provided regarding the MAC strain used to infect the animals. It is stated that it is a clinical isolate. Why wasn’t a known strain of MAC used to allow comparison to other published studies? What were the clinical characteristics of the individual the isolate was obtained from, did that person have cavitary pulmonary MAC, were they transiently colonized etc.?

3) How does this model differentiate between the relatively self- limited condition of hypersensitivity pneumonitis which is observed in humans exposed to water in contaminated hot tubes compared to cavitary or fibronodular MAC in humans? The radiograph of marmoset lung looks and is reported as pneumonitis, not cavitary or nodular disease.

4) There is no description of how lung histopathology was assessed. Did a veterinary pathologist review the slides. Scanning morphometry would be the appropriate way to measure lung airways and make a determination that “early bronchiectasis”. Without this, no definitive conclusions can be drawn regarding the occurrence of bronchiectasis in this model.

5) The data suggest a self-limited resolving infection comparing D30 to D60. This does not mimic chronic progressive infection in humans.

6) It is not surprising that serum and lung cytokine levels are elevated in infected animals with an endobronchial instillation of 1E8 CFU. It is hard to draw any specific conclusions from the cytokine data.

7) The data does not support the following statement in the Discussion section – “The animals showed no discernable clinical signs of lung infection, such as cough, decreased activity or weight loss, suggesting that the inoculation resulted in a “sub-clinical” chronic infection.” Clinical infection in humans is accompanied by cough and weight loss. Without actual lung CFU data and without evidence progressive inflammation comparing D30 to D60 animals the data are not convincing for chronic infection as is seen in human pulmonary MAC infection.

8) The data does not support the following statement in the discussion – “This type of indolent infection is consistent with the nodular/bronchiectasis form of M. intracellulare lung disease in humans”. As stated above the evidence for chronic infection is not convincing, and the presented radiographic finding of “pneumonitis” is not consistent with chronic pulmonary MAC infection in humans.

6. PLOS authors have the option to publish the peer review history of their article (what does this mean?). If published, this will include your full peer review and any attached files.

Reviewer #1: No

Reviewer #2: No

---

## [Author Response · Author response to Decision Letter 0]

18 Feb 2022

We would like to thank the reviewers for their time and efforts in revising our work. We believe that their comments further strengthen our manuscript. We have responded point-by-point to their comments. 

Reviewer #1: In this manuscript by Peters, et al., the authors seek to fill a major hole in MAC research- the lack of a tractable animal model. Marmosets have been used recently to model Mtb infection and appear to be quite susceptible to Mtb. Here, authors infected 7 marmosets bronchoscopically with high-dose M. intracellulare and 2/3 were culture-positive at necropsy 30 dpi; 3/4 at necropsy 60 dpi. BAL was culture-negative in all cases. There were signs of pneumonitis and some histopathology in the 5 infected animals, although all appeared clinically normal. Culture-positive animals exhibited higher cytokine levels in BALF and serum than did the uninfected animals. M. intracellulare resulted in a mild disease course in marmosets, consistent with MAC presentation in most humans. One issue with this study is that CXR is quite insensitive for detecting subtle changes in lungs and CT imaging would have much higher resolution. However, the authors do note this in the Discussion and the careful histopathology examination was capable of characterizing lung pathology.

The manuscript from leaders in the MAC field is well-written and the studies were well-designed and executed. Though done with a small number of animals (only 5 animals were apparently infected), this report does demonstrate some utility of the marmoset model and may be useful for future studies of MAC. One downside to the model is the inconsistency in establishing infection.

The existing manuscript has no flaws that should preclude publication as-is. Only one minor concern lingers: Cytokines were measured with an Invitrogen 28-plex panel. Such multiplex panels have variable ability to detect cyto/chemokines from various NHP species. Was the ability to detect marmoset cyto/chemokines validated?

Response to Reviewer 1: Luis Giavedoni PhD, is the Senior Professor for the Biology Core for our Primate Center [Texas Biomedical Research Institute] and has validate both cytokines and chemokines for marmosets. The reviewer is correct in this comment since only a fraction of marmoset cytokines are detected with certain human-specific reagents. The M&M section pertaining to cytokine detection with the Luminex system has been modified as follows: The inflammatory environment was assessed for all of the subjects by evaluating circulating cytokine concentrations measured using the Luminex system as validated for marmosets and other nonhuman primates (Giavedoni, 2005; Höglind et al., 2017; Ross et al., 2019). The assay included evaluation of the following 18 analytes: interferon alpha (IFN-alpha), interferon gamma (IFN-γ), interleukin-1 beta (IL-1β), IL-1 receptor antagonist (IL-1RA), IL-2, IL-4, IL-7, IL-12 p40, IL-18, IL-23 monocyte chemoattractant protein 1 (MCP-1, CCL2), macrophage migration inhibitory factor (MIF), monokine induced by gamma interferon (MIG, CXCL9), macrophage inflammatory protein 1-alpha (MIP-1α, CCL3), MIP-1b (CCL4), regulated on activation, normal T cell expressed and secreted (RANTES, CCL5), tumor necrosis factor-alpha (TNF-α), and soluble intercellular adhesion molecule 1 (sICAM-1). Cytokine concentrations for infected animals were evaluated with multivariate ANOVA.

Giavedoni LD. Simultaneous detection of multiple cytokines and chemokines from nonhuman primates using luminex technology. J Immunol Methods. 2005 Jun;301(1-2):89-101.

Höglind A, Areström I, Ehrnfelt C, Masjedi K, Zuber B, Giavedoni L, Ahlborg N. Systematic evaluation of monoclonal antibodies and immunoassays for the detection of Interferon-γ and Interleukin-2 in old and new world non-human primates. J Immunol Methods. 2017 Feb;441:39-48.

Ross CN, Adams J, Gonzalez O, Dick E, Giavedoni L, Hodara VL, Phillips K, Rigodanzo AD, Kasinath B, Tardif SD. Cross-sectional comparison of health-span phenotypes in young versus geriatric marmosets. Am J Primatol. 2019 Feb;81(2):e22952.

Reviewer #2: There are a number of questions and concerns regarding the manuscript:

1) Although it is recognized that the number of animals used in the study was limited, it is not possible to draw conclusions regarding the presence of productive/progressive lung infection in the infected animals based on lung CFU data. The endobronchial instillation of 1E8 CFU is very high and much higher than what a human would be exposed to in the natural environment. Interpretation is complicated by the lack of lung deposition CFU after inoculation which is standard in lung infection experiments. Finally, the choice of scoring positive cultures as countable colonies rather than a more standard criteria such as CFU per gram of lung tissue further clouds the interpretation. Within the countable colony from 1-49 the range is quite large, if 1 or 2 colonies were present this would represent negligible recovery of bacteria from the lung.

Response 1: The inoculum of Mycobacterium avium complex was chosen based on an inoculation dose causing MAC disease in rhesus macaques (Am J Respir Cell Mol Biol; 54(2):170-176). This study was not intended to investigate a dose-response threshold for establishing MAC infection in marmosets. That type of study would require a large number of animals and enormous expense. Rather, the goal of this project was simply to determine if establishing pulmonary MAC infection was possible in the marmoset. Choosing an inoculation dose shown to cause pulmonary disease in another primate model appeared to offer the best chances of success with the least number of animals. To the reviewer’s point about MAC inoculum in human disease, we submit that inoculum is unknown. Likely for humans, the size of the inoculum is not as important as the duration of the exposure. Additionally, in humans, structural lung disease such as bronchiectasis appears to be necessary for MAC infection in the majority of cases so that it is not surprising in an animal with a normal tracheobronchial tree a relatively large inoculum would be required to establish infection. It is also noteworthy that most current animal models for MAC disease require some systemic immune suppression or interruption of airway defenses or architecture. In that regard, the marmoset model offers the opportunity to assess MAC infection in an uncomplicated way.

2) Minimal information is provided regarding the MAC strain used to infect the animals. It is stated that it is a clinical isolate. Why wasn’t a known strain of MAC used to allow comparison to other published studies? What were the clinical characteristics of the individual the isolate was obtained from, did that person have cavitary pulmonary MAC, were they transiently colonized etc.?

Response 2: We chose this MAC strain because it was both macrolide and amikacin resistant. The chances of finding a “wild-type” MAC isolate with both macrolide and amikacin resistance is vanishingly small. Having this in vitro susceptibility pattern eliminated the need (and expense) for genotyping isolates recovered from the animals in comparison to the instilled MAC isolate. There was no doubt that the infecting organism was the organism that had been instilled. The isolate came from a cystic fibrosis patient with extensive MAC infection who had failed therapy with macrolide and amikacin containing regimens after developing macrolide and amikacin resistance. We knew, therefore, that it was a pathogenic isolate and that it could be easily identified. 

3) How does this model differentiate between the relatively self- limited condition of hypersensitivity pneumonitis which is observed in humans exposed to water in contaminated hot tubes compared to cavitary or fibronodular MAC in humans? The radiograph of marmoset lung looks and is reported as pneumonitis, not cavitary or nodular disease.

Response 3: Hypersensitivity MAC infection in humans is a diffuse process radiographically associated with the acute onset of symptoms (cough, dyspnea). None of the animals were symptomatic and none had radiographic findings consistent with hypersensitivity pneumonitis. We agree that the animals appeared to have an acute focal process consistent with pneumonitis. We submit that both fibrocavitary disease and nodular/bronchiectatic disease are chronic manifestations of MAC lung disease. Early or primary events in human MAC lung disease are unknown. There is no model of the form fruste of MAC lung disease. It is not known, for instance if there is latent MAC infection similar to TB. To determine if over time marmoset develops fibrocavitary disease or nodular/bronchiectatic disease would require a much longer study, although some exhibited bronchiectasis even at 60 days. We think the marmoset could be an appropriate animal model to make that determination for MAC lung disease but that was not in the scope of this initial trial to determine if infection could be established. 

4) There is no description of how lung histopathology was assessed. Did a veterinary pathologist review the slides. Scanning morphometry would be the appropriate way to measure lung airways and make a determination that “early bronchiectasis”. Without this, no definitive conclusions can be drawn regarding the occurrence of bronchiectasis in this model.

Response 4: The lung tissue was examined by Dr. Jackie Colson, an internationally recognized PhD in Pulmonary Pathology. She is recognized as an expert in primate pathology with over 50 publications on lung injury in non-human primates. The tissue was processed by in coronal sections to allow better visualization of airways and morphometry. Each lobe was read independently by Dr. Coalson and reviewed in our pulmonary pathology research conference. The definition of bronchiectasis was the lack of airways reduction in diameter and the extension of the dilated airway extending to the pleural surface or within 1 mm of the pleura. We chose the term “early bronchiectasis” rather than “persistent bronchiectasis” since we only followed the animals out to 60 days.

Examples of a few of Dr. Coalson’s nonhuman primate articles include:

Yoder BA, Coalson JJ. Animal models of bronchopulmonary dysplasia. The preterm baboon models. Am J Physiol Lung Cell Mol Physiol. 2014 Dec 15;307(12):L970-7.

Coalson JJ, Winter V, deLemos RA. Decreased alveolarization in baboon survivors with bronchopulmonary dysplasia. Am J Respir Crit Care Med. 1995 Aug;152(2):640-6. 

Coalson JJ, Winter VT, Gerstmann DR, Idell S, King RJ, Delemos RA. Pathophysiologic, morphometric, and biochemical studies of the premature baboon with bronchopulmonary dysplasia. Am Rev Respir Dis. 1992 Apr;145(4 Pt 1):872-81.

de los Santos R, Coalson JJ, Holcomb JR, Johanson WG Jr. Hyperoxia exposure in mechanically ventilated primates with and without previous lung injury. Exp Lung Res. 1985;9(3-4):255-75.

5) The data suggest a self-limited resolving infection comparing D30 to D60. This does not 

mimic chronic progressive infection in humans.

Response 5: As discussed under #3, it is incompletely understood what the manifestations (clinical, microbiologic and radiographic) of primary MAC infection in humans are. The infected animals did lose weight during the first 2 weeks of the study and then tended to return toward baseline even though the intake of food and water was maintained. This point was added to the manuscript and reflected in the data presented in Table 1. We did not follow the animals long enough to determine if the infections would completely resolve or transiently improve and then recur (similar to primary TB) or if the animals would develop a latent infection (similar to latent TB infection). It is even possible that the marmoset response we observed is applicable to human MAC lung disease, which is why developing animal models is so important. There is simply not enough known about early MAC infection in humans to make definitive conclusions. 

6) It is not surprising that serum and lung cytokine levels are elevated in infected animals with an endobronchial instillation of 1E8 CFU. It is hard to draw any specific conclusions from the cytokine data.

Response 6: The reviewer raises a valid point. Yet, we do not think it is very plausible that the MAC instillation per se caused the cytokine elevations. The MAC instillation was associated with an acute inflammatory event in the lung consistent with pneumonitis due to MAC infection. In two animals, we instilled the same number of CFU of M. Abscessus and found no cytokine response and normal histopathology 30 days after inoculation. The animals receiving MAC clearly had granulomatous inflammation and pneumonitis on pathologic exam and the cytokines remained elevated in the BAL at day 60.

7) The data does not support the following statement in the Discussion section – “The animals showed no discernable clinical signs of lung infection, such as cough, decreased activity or weight loss, suggesting that the inoculation resulted in a “sub-clinical” chronic infection.” Clinical infection in humans is accompanied by cough and weight loss. Without actual lung CFU data and without evidence progressive inflammation comparing D30 to D60 animals the data are not convincing for chronic infection as is seen in human pulmonary MAC infection.

Response 7: We agree with the reviewer. The word “chronic” has been deleted. We agree with the reviewer that with the comment that the observation that the infection was “sub-clinical”. 

8) The data does not support the following statement in the discussion – “This type of indolent infection is consistent with the nodular/bronchiectasis form of M. intracellulare lung disease in humans”. As stated above the evidence for chronic infection is not convincing, and the presented radiographic finding of “pneumonitis” is not consistent with chronic pulmonary MAC infection in humans.

Response 8: We agree with the Reviewer’s comment and we have corrected the manuscript accordingly and removed the statement.

---

## [Decision Letter · Decision Letter 1]

9 Jun 2022

PONE-D-21-34410R1A Marmoset Model for Mycobacterium avium Complex Pulmonary DiseasePLOS ONE

Dear Dr. Peters,

Thank you for submitting your manuscript to PLOS ONE. After careful consideration, we feel that it has merit but does not fully meet PLOS ONE’s publication criteria as it currently stands. Therefore, we invite you to submit a revised version of the manuscript that addresses the points raised during the review process.

ACADEMIC EDITOR: Though the authors have elaborately justified their views to reviewer comments, Several key points, related to limitations of the study, are missing in the revised manuscript itself. The authors should incorporate their response to reviewers into the discussion section that deals with limitations of the study. For example, the following should be mentioned: The inoculum of Mycobacterium avium complex was chosen based on an inoculation dose causing MAC disease in rhesus macaques (Am J Respir Cell Mol Biol; 54(2):170-176). This study was not intended to investigate a dose-response threshold for establishing MAC infection in marmosets. That type of study would require a large number of animals and enormous expense. Rather, the goal of this project was simply to determine if establishing pulmonary MAC infection was possible in the marmoset. Choosing an inoculation dose shown to cause pulmonary disease in another primate model appeared to offer the best chances of success with the least number of animals. To the reviewer’s point about MAC inoculum in human disease, we submit that inoculum is unknown. Likely for humans, the size of the inoculum is not as important as the duration of the exposure. Additionally, in humans, structural lung disease such as bronchiectasis appears to be necessary for MAC infection in the majority of cases so that it is not surprising in an animal with a normal tracheobronchial tree a relatively large inoculum would be required to establish infection. It is also noteworthy that most current animal models for MAC disease require some systemic immune suppression or interruption of airway defenses or architecture. In that regard, the marmoset model offers the opportunity to assess MAC infection in an uncomplicated way.Similarly, the number of animals per timepoint (n=3-4) and usage of one sex (female) are confounding factors for proper statistical analysis. This should be mentioned in the discussion.

Revising the manuscript with these point would address the concern of the new reviewer comments posted below. 

We look forward to receiving your revised manuscript.

Kind regards,

Selvakumar Subbian, Ph.D.

Academic Editor

PLOS ONE

Journal Requirements:

Reviewers' comments:

Reviewer's Responses to Questions

**Comments to the Author**

1. If the authors have adequately addressed your comments raised in a previous round of review and you feel that this manuscript is now acceptable for publication, you may indicate that here to bypass the “Comments to the Author” section, enter your conflict of interest statement in the “Confidential to Editor” section, and submit your "Accept" recommendation.

Reviewer #3: (No Response)

2. Is the manuscript technically sound, and do the data support the conclusions?

Reviewer #3: No

3. Has the statistical analysis been performed appropriately and rigorously? 

Reviewer #3: No

4. Have the authors made all data underlying the findings in their manuscript fully available?

Reviewer #3: Yes

5. Is the manuscript presented in an intelligible fashion and written in standard English?

Reviewer #3: Yes

6. Review Comments to the Author

Reviewer #3: Authors has to carried out a marmoset model of animal experiment to study the pathogenesis of nontuberculous mycobacterial disease. After infection, authors monitored several disease parameters including bacterial burden in several organs, cytokine levels and histopathology at 30 and 60 days of post infection. However, this study has to face the following concerns.

Major comments:

Even though all marmosets were infected with high dose of inoculum (10E8 of MAC) by endobronchial inoculation, infection was established in only 5 animals out 7 marmosets. This result indicate that more than 25% animals did not get infection. Has this study enough statistical power to bring any conclusion?

The authors concluded that this study would be useful to recapitulate M. avium complex lung infection in humans. However, proper standardization of inoculum should be carried out to establish infection in all infected marmosets with minimum standard deviation.

7. PLOS authors have the option to publish the peer review history of their article (what does this mean?). If published, this will include your full peer review and any attached files.

Reviewer #3: No

---

## [Author Response · Author response to Decision Letter 1]

22 Aug 2022

See attached PDF with detailed responses

---

## [Decision Letter · Decision Letter 2]

20 Sep 2022

PONE-D-21-34410R2A Marmoset Model for Mycobacterium avium Complex Pulmonary DiseasePLOS ONE

Dear Dr. Peters,

Thank you for submitting your manuscript to PLOS ONE. After careful consideration, we feel that it has merit but does not fully meet PLOS ONE’s publication criteria as it currently stands. Therefore, we invite you to submit a revised version of the manuscript that addresses the points raised during the review process.

ACADEMIC EDITOR: Please address the following: 

Table-1. The numbering (superscripts and non-superscript) is very confusing and hard to understand. Please re-organize the descriptions.

Figure-1. Scale bar and magnification are missing; please add this info on the image and/or mention in the legend.

Figure-3. Specify what the arrow denotes?. Scale bar and magnification are missing; please add this info on the image and/or mention in the legend.

We look forward to receiving your revised manuscript.

Kind regards,

Selvakumar Subbian, Ph.D.

Academic Editor

PLOS ONE

Journal Requirements:

Reviewers' comments:

Reviewer's Responses to Questions

**Comments to the Author**

1. If the authors have adequately addressed your comments raised in a previous round of review and you feel that this manuscript is now acceptable for publication, you may indicate that here to bypass the “Comments to the Author” section, enter your conflict of interest statement in the “Confidential to Editor” section, and submit your "Accept" recommendation.

Reviewer #3: All comments have been addressed

2. Is the manuscript technically sound, and do the data support the conclusions?

Reviewer #3: Partly

3. Has the statistical analysis been performed appropriately and rigorously? 

Reviewer #3: I Don't Know

4. Have the authors made all data underlying the findings in their manuscript fully available?

Reviewer #3: Yes

5. Is the manuscript presented in an intelligible fashion and written in standard English?

Reviewer #3: Yes

6. Review Comments to the Author

Reviewer #3: (No Response)

7. PLOS authors have the option to publish the peer review history of their article (what does this mean?). If published, this will include your full peer review and any attached files.

Reviewer #3: No

---

## [Author Response · Author response to Decision Letter 2]

29 Nov 2022

Reviewer #1 suggested publication but questioned if the 28 plex used for this study had been validated. We pointed out that this 28 Plex cytokine/chemokine assay was validated by one of our authors and submitted the publication for review. This PLEX has been used by the Texas Biomedical Research institute for all of our NIH studies over the past 20 years.

Reviewer #2 questioned if the pathologist that reviewed the marmoset histology had prior experience with non-human primates pathology. We submitted over 15 articles written by Dr. Jacqueline Coalson: a PhD pulmonary pathologist who reviewed all our NIH ARDS non-human primate studies (an 8 year Program Project). She has also been the pulmonary pathologist reviewing all the neonatal baboon studies performed at our institution.The reviewer also stated that using such a high dose of MAC may have accounted for why we observed the pathological changes at necropsy; however, we instilled the same concentration of organism with M. abscessus and had no radiological or pathological changes in 3 marmosets. The reviewer also questioned our use of the term "early bronchiectasis" and we explained that even though the airway extended to the pleura that we only followed the marmoset for 60 days. Therefore, we could not prove these findings would persist over a longer period of observation [thus we used the term "early bronchiectasis"].

The 3rd reviewer stated he was unable to determine if the study was scientifically sound. We had no statistical analysis to support our observations. We asked the statistician at National Jewish Medical Center to independently review the cytokine data and determine if it was statistically significant and supported our observations. He concluded that the serial cytokine data that was obtained at baseline and on a weekly basis were statistically significant. We feel this analysis significantly strengthens our submission. Additionally, only one other study in the medical literature has demonstrated the ability to reproduce MAC in a non-human primate model (in 1 out of 3 animals). We feel all the questions from the reviewers have been adequately addressed and feel this manuscript adds important information to the medical literature.

---

## [Editor Report · Decision Letter 3]

1 Dec 2022

A Marmoset Model for Mycobacterium avium Complex Pulmonary Disease

PONE-D-21-34410R3

Dear Dr. Peters,

We’re pleased to inform you that your manuscript has been judged scientifically suitable for publication and will be formally accepted for publication once it meets all outstanding technical requirements.

Kind regards,

Selvakumar Subbian, Ph.D.

Academic Editor

PLOS ONE
---

## [Editor Report · Acceptance letter]

21 Dec 2022

PONE-D-21-34410R3 

A Marmoset Model for *Mycobacterium avium* Complex Pulmonary Disease 

Dear Dr. Peters:

I'm pleased to inform you that your manuscript has been deemed suitable for publication in PLOS ONE. Congratulations! Your manuscript is now with our production department. 

Kind regards, 

on behalf of

Dr. Selvakumar Subbian 

Academic Editor

PLOS ONE